# Patterns of prescription dispensation and over-the-counter medication sales in Sweden during the COVID-19 pandemic

**Pär Karlsson[1], Aya Olivia Nakitanda[1], Lukas Löfling[1,2], Carolyn E. Cesta[1] ***

**1** Centre for Pharmacoepidemiology, Karolinska Institutet, Stockholm, Sweden, **2** Department of Research, Etiological Research Unit, Cancer Registry of Norway, Oslo, Norway

* carolyn.cesta@ki.se

## Abstract

### Introduction

On February 26th 2020, a high alert was issued in Sweden in response to the diagnosis of the first few coronavirus disease 2019 (COVID-19) cases in the country. Subsequently, a decreased supply of essential goods, including medical products, was anticipated. We aimed to explore the weekly patterns of prescription dispensing and over-the-counter (OTC) medication sales in Sweden in 2020 compared with previous years, to assess the influence of the government restrictions on medication sales, and to assess whether there is evidence of medication stockpiling in the population.

### Methods

Aggregated data on the weekly volume of defined daily doses (DDDs) of prescription medication dispensed and OTC sales from 2015 to 2020 were examined. From 2015–2019 data, the predicted weekly volume of DDDs for 2020 was estimated and compared to the observed volume for each ATC anatomical main group and therapeutic subgroup.

### Results

From mid-February to mid-March 2020, there were increases in the weekly volumes of dispensed medication, peaking in the second week of March with a 46% increase in the observed versus predicted number of DDDs dispensed (16,440 vs 11,260 DDDs per 1000 inhabitants). A similar pattern was found in all age groups, in both sexes, and across metropolitan and non-metropolitan regions. In the same week in March, there was a 96% increase in the volume of OTC sold (2,504 vs 1,277 DDDs per 1000 inhabitants), specifically in ATC therapeutic subgroups including vitamins, antipyretics, painkillers, and nasal, throat, cough and cold preparations.

### Conclusion

Beginning in mid-February 2020, there were significant changes in the volume of prescription medication dispensed and OTC drugs sold. The weekly volume of DDDs quickly

**Data Availability Statement:** We are not able to share the data used in this study as it was received from third parties. This descriptive study is based on aggregated level data from the population-

based health registers in Sweden and can be requested from Sweden's National Board of Health and Welfare (https://www.socialstyrelsen.se/statistik-och-data/bestalla-data-och-statistik/) and the Swedish eHealth Agency (https://www.ehalsomyndigheten.se/om-oss/lakemedelsstatistik/).

**Funding:** The author's research group at the Centre for Pharmacoepidemiology, Karolinska Institutet, provided financial support for this project.

**Competing interests:** The authors have declared that no competing interests exist.

decreased following recommendations from public authorities. Overall, our findings suggest stockpiling behavior over a surge in new users of medication.

## Introduction

While the outbreak of severe acute respiratory syndrome coronavirus 2 (SARS-CoV-2) became classified as a public health emergency of international concern by the World Health Organization (WHO) on December 30[th] 2020, the first confirmed individual infected with SARS-CoV-2 was not detected in Sweden until January 31, 2020 [1,2]. On February 26[th], the Swedish National Board of Health and Welfare (*Socialstyrelsen* in Swedish) issued a high alert in Sweden, and in subsequent weeks social restrictions and a decreased supply of essential goods, including medical products, were anticipated based on the experience in other European countries. As a preventive action, the Swedish Medical Products Agency (*Läkemedelsverket* in Swedish) first recommended that pharmacies restrict the amount of prescription medication dispensed and over-the-counter (OTC) medication sold to individuals on March 19[th], and then mandated the restriction on April 1[st], 2020 [3,4]. In Sweden, prescriptions for long term treatments are commonly valid for one year at a time with four refills. The government mandate restricted pharmacies to dispense a maximum of 90 days' supply of prescription medication at a time, and only if two-thirds of the previous dispensation had been consumed. Hence, prior to April 1[st] an individual could, if they were willing to pay the full cost of the medication, dispense as much as their prescription allowed.

Implementation of these restrictions indicated that excessive medication purchasing and stockpiling were thought to be occurring in Sweden during this period, but its extent has not been quantified. In Sweden, as well as in neighboring Finland and Norway, the public health institutions have reported that an increase in dispensations and OTC medication sales occurred during the weeks in March 2020 compared to the same period in 2019 [5–7].

We therefore aimed to explore the weekly patterns of prescription dispensing and OTC medication sales in Sweden from the start of 2020 compared to patterns predicted from data from the previous 5 years. Further, we aimed to assess the influence of the government restrictions on medication sales and whether there is evidence of medication stockpiling by individuals in the population.

## Materials and methods

### Data sources

Aggregated data on pharmacy dispensations of prescription medications from January 1, 2015 to December 29, 2020 were collected from the Swedish National Board of Health and Welfare. This included the volume, measured in defined daily doses (DDD), dispensed per week for all anatomical main groups of the Anatomical Therapeutic Chemical (ATC) Classification System, except group V, and corresponding therapeutic subgroups (Table 1). Group V consists of many types of products without assigned DDDs [8]. Data were received for the total population and stratified by sex, age group (0–19, 20–39, 40–59, 60–79, 80+ years), and type of geographical region (metropolitan: counties containing Sweden's three largest cities, non-metropolitan: all other counties).

Similarly, total daily sales of OTC medication in DDDs from pharmacies and grocery stores were requested from the Swedish eHealth Agency (*E-hälsomyndigheten* in Swedish) for the period January 1, 2015 to December 29, 2020, for all ATC anatomical main groups and

**Table 1. Anatomical main groups of Anatomical Therapeutic Chemical (ATC) classification system included in the aggregated data for prescription dispensations and over-the-counter medication sales.**

| ATC group | Description | Prescription data | OTC data |
|:---:|---|:---:|:---:|
| A | Alimentary tract and metabolism | ✓ | ✓ |
| B | Blood and blood forming organs | ✓ | ✓ |
| C | Cardiovascular system | ✓ | |
| D | Dermatologicals | ✓ | |
| G | Genito urinary system and sex hormones | ✓ | ✓ |
| H | Systemic hormonal preparations, excl. sex hormones and insulins | ✓ | |
| J | Antiinfectives for systemic use | ✓ | |
| L | Antineoplastic and immunomodulating agents | ✓ | |
| M | Musculo-skeletal system | ✓ | ✓ |
| N | Nervous system | ✓ | ✓ |
| P | Antiparasitic products, insecticides and repellents | ✓ | ✓ |
| R | Respiratory system | ✓ | ✓ |
| S | Sensory organs | ✓ | |

therapeutic subgroups, except group V (Table 1). Many medications are not available OTC, hence not every ATC anatomical main group is represented in the data. Daily DDDs sold were then summed into weekly sales. Characteristics of the purchasers are not available for these OTC sales data.

For both data sources, weeks were started on a Wednesday to assess the impact of the Swedish Medical Products Agency ordinance issued on Wednesday April 1st, 2020. The total population of Sweden per year and per strata was obtained from the Statistics Sweden website (downloaded 2021-03-12) [9].

## Analysis

The number of DDDs dispensed or sold per 1000 individuals in the Swedish population per week were calculated. In Sweden, the Easter holiday is a week-long vacation for school-aged children and is commonly a time for family travel. Since Easter occurs during different weeks each year, a direct comparison of weeks in 2020 to the same weeks in 2015–2019 is suboptimal. Therefore, an Analysis of Covariance (ANCOVA) model was used to develop a prediction model for the expected volume of DDDs dispensed or sold in 2020 based on data from 2015 to 2019. The model included: 1) the date as a continuous variable to model linear change of drug use over time, 2) a categorical week variable to model seasonal patterns, 3) a categorical Easter variable to model the Easter holidays (including four categories: the week before Easter, the week of Easter, the week after Easter, and other weeks), and finally 4) a continuous variable for number of working days (including Saturdays) in each week to model other holidays than Easter. This last element was included because in many non-metropolitan regions in Sweden, pharmacies are not open on Sundays or on holidays. Observations for each week in 2020 were then compared to the predicted value, including 95% confidence limits (CLs) for individual predictions, and a ratio of observed to predicted values was calculated. In the figures, multiplicity concerning the many weekly comparisons within a drug group, but not across drug groups, is applied by dividing the standard significance limits by 52 (i.e, Bonferroni correction based on the number of weeks of data in 2020). In order to give the readers the possibility to apply any multiplicity correction they find appropriate, the p-values are presented uncorrected for multiple comparisons in S1 and S2 Tables.

All calculations were performed in SAS version 9.4 and SAS/STAT 14.3.

## Results

During the first week of March in 2020, the overall volume of DDDs of prescription medication dispensed (all ATC anatomical main groups) started to increase above the amount predicted from the weekly data of the preceding 5 years, and reached a peak corresponding to a 46% increase (observed: 16.440 DDDs per 1000 inhabitants vs. predicted: 11,260 DDDs per 1000 inhabitants (95%CL 10,527 to 11,993) in the week of March 11[th] to 17[th] 2020. The number of DDDs dispensed subsequently decreased the following week (March 18[th] to 24[th]) yet remained 14% above the predicted amount, until the last week of March when dispensed DDDs returned to the predicted pattern which was maintained throughout the remainder of the available data for 2020 (Fig 1a and S1 Table).

The surge in the number of DDDs dispensed in March 2020 occurred across all age groups (Fig 1b). Notably, the 0–19 year age group had the highest peak, corresponding to a 55% increase in the number of DDDs dispensed during the week of March 11[th] to 17[th] 2020 (observed: 2,858 DDDs per 1000 inhabitants vs. predicted: 1,845 DDDs per 1000 inhabitants (95%CL 1,685 to 2,004). Similarly, in the week of March 11[th] there was a significant increase in the number of DDDs dispensed for both women (48% increase) and men (43% increase), and for both metropolitan (47% increase) and non-metropolitan (45% increase) regions (S1a and S1b Fig).

For all ATC anatomical main groups, there was a peak in the number of DDDs dispensed during the week starting on March 11[th] with a statistically significant increase between the observed versus predicted amount (Fig 2). During this week, ATC group R and P had the largest increase with 75% more DDDs dispensed than predicted. For ATC group R, this increase was primarily driven by the 127% increase in DDDs for drugs in the R03 therapeutic subgroup (*Drugs for obstructive airway diseases*), and a significantly increased amount of DDDs dispensed was sustained over the remaining weeks of March. For ATC group P, in the following week starting March 18[th], there was an even larger increase in DDDs dispensed, 123% more than predicted. However, the overall the volume of DDDs dispensed per week during the study period is very low (max 9.3 DDDs per 1000 inhabitants) and there was an increase in the amount of DDDs dispensed in 2018 and 2019 compared to the prior 3 years, resulting in a poor model fit and consequently a predicted value for 2020 with larger CLs (S2 Fig).

Notably, while ATC group J had a small yet significant increase (13%) in DDDs dispensed in the week of March 11[th], during the weeks of April, May, and June there were significantly fewer DDDs dispensed (15–19% less) than predicted for 2020, and remained low for the remainder of the year. Specifically for therapeutic subgroup J01 (*antibacterials for systemic use*), there was a statistically significant reduction in the number of DDDs dispensed in almost every week starting at the beginning of April until the end of the available data in December 2020.

OTC sales in 2020 followed similar trends as dispensed medication (Fig 3a). Beginning in the last week of February, there was an increase in the DDDs sold every week until the last week of March, with a 96% increase in DDDs sold in the week beginning with March 11[th] (observed: 2,504 DDDs per 1000 inhabitants vs. predicted: 1,277 DDDs per 1000 inhabitants (95%CL 1,148 to 1,407). After the week starting April 1[st], there was no difference between the observed and predicted sales in any of the weeks, except for a 17% reduction in OTC DDDs sold in the last week of May (S2 Table). Fig 3b shows the OTC DDDs sold for the six ATC anatomical main groups which had statistically significant increases between the observed versus predicted weekly sales at least one week in March 2020 (ATC groups A, B, G, M, N, R).

ATC anatomical main groups A and B had a 62% and 34% increase in observed versus predicted sales of OTC DDDs in the week starting with March 11[th], respectively. This increase

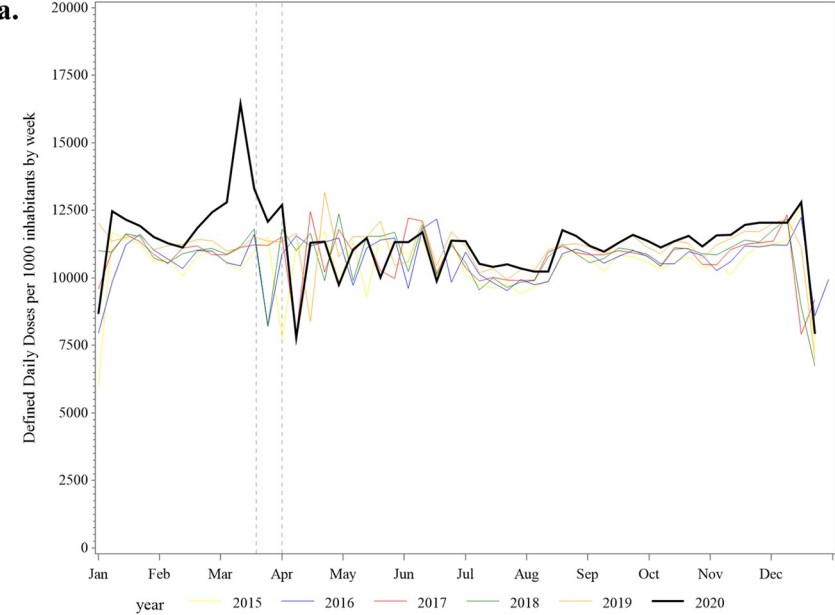

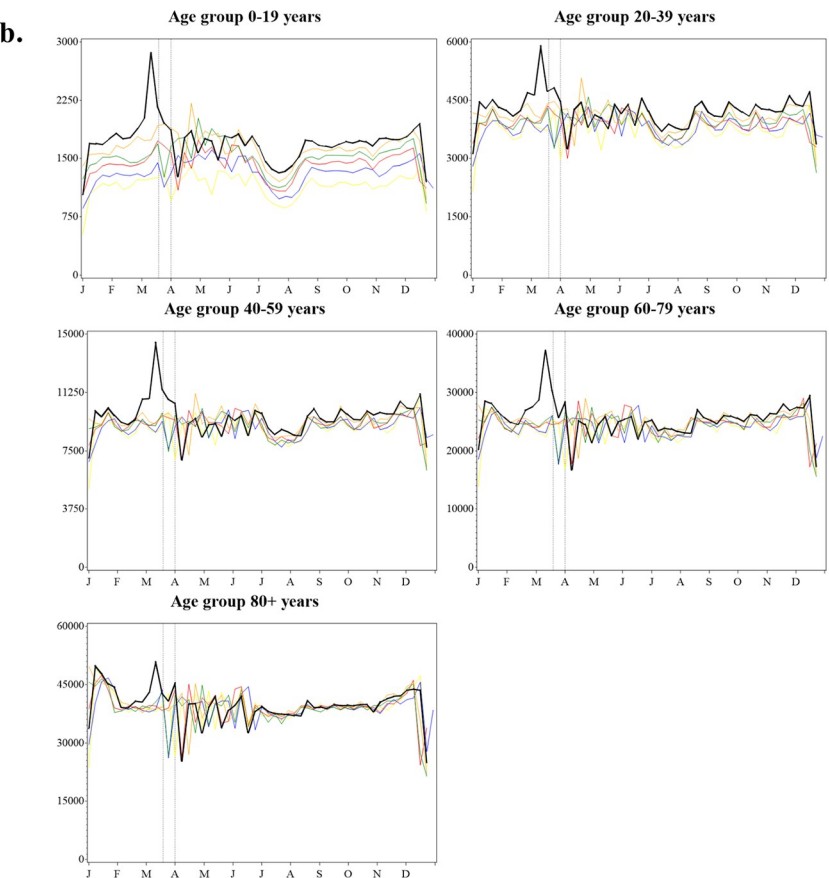

**Fig 1. Volume of dispensed defined daily doses of prescription medication per 1000 inhabitants by week, 2015–2020, Sweden (a) the entire population and (b) by age group.** Note: The vertical lines indicate the weeks containing March 19[th] and April 1[st] 2020 when limits on medication sales were recommended and then mandated in Sweden, respectively.

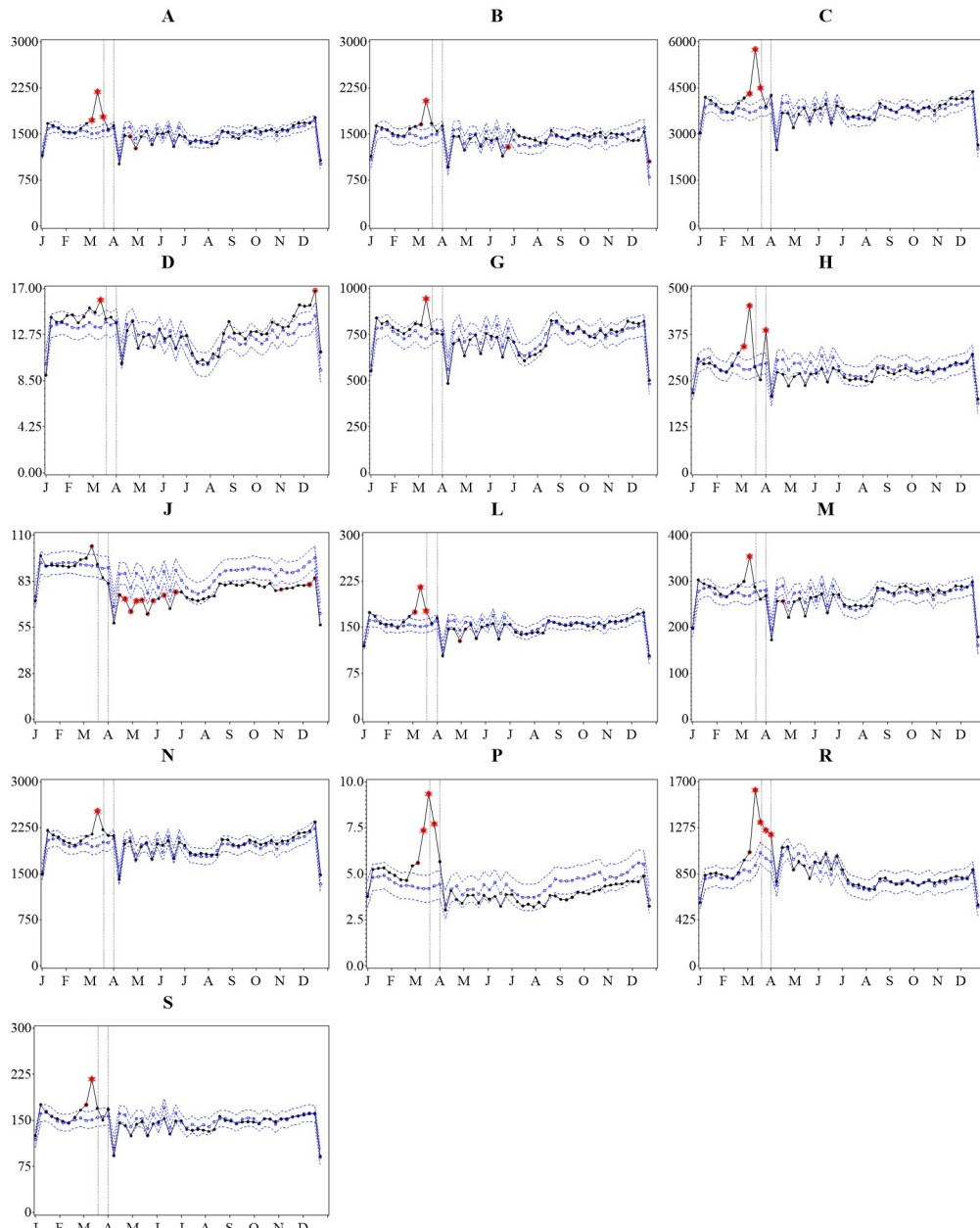

**Fig 2. Observed versus predicted *volume* of dispensed defined daily doses of prescription medications per 1000 inhabitants by ATC anatomical main group, 2020, Sweden.** Note: The vertical lines indicate the weeks containing March 19th and April 1st 2020 when limits on medication sales were recommended and then mandated, respectively. P-values that remain significant after Bonferroni correction are marked. Legend: solid black line = observed values; dotted blue line with open circles = predicted values; dotted blue lines, no marker = upper and lower 95% confidence limits; single red circle = 5% significance level; double red circle = 1% significance level; double red circle with star = 0.1% significance level.

was driven primarily by medications in the therapeutic subgroups A11 (382% increase, *Vitamins*), and B03 (34% increase, *antianemic preparations*). Both ATC A and B groups had subsequent notable reductions in observed sales in DDDs later in the year (A: -15 to -25% in weeks in May, June, August, September and October; B: -21 to -72% in all weeks in May and June,

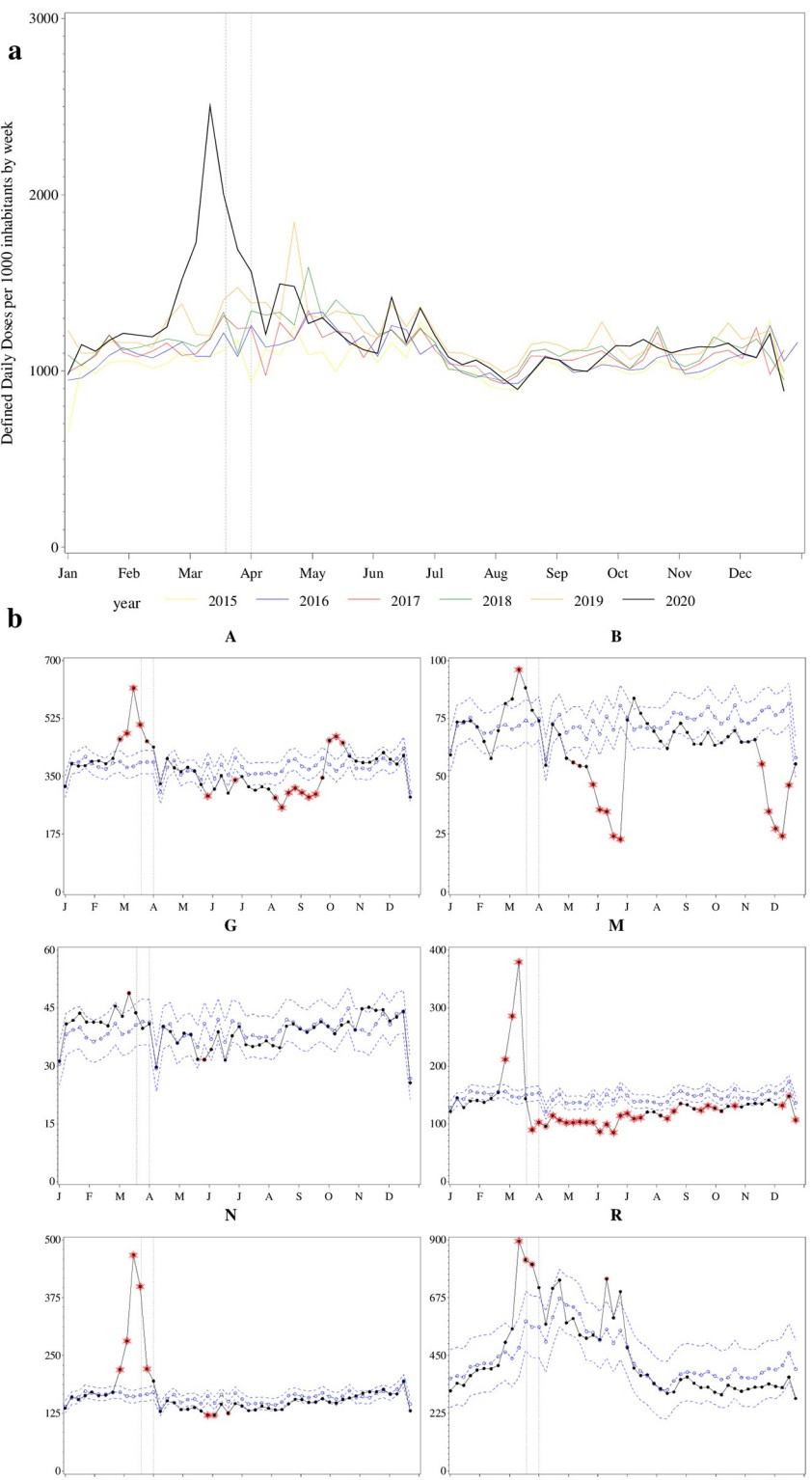

**Fig 3. (a) Volume of defined daily doses sold of over-the-counter medication per 1000 inhabitants by week, 2015–2020, Sweden; (b) Observed versus predicted weekly volume of defined daily doses sold of over-the-counter medication per 10000 inhabitants by ATC anatomical main group, 2020, Sweden.** Note: The vertical lines indicate the weeks containing March 19th and April 1st 2020 when limits on medication sales were recommended and then mandated, respectively. P-values that remain significant after Bonferroni correction are marked. Legend: <u>solid black</u>

line = observed values; dotted blue line with open circles = predicted values; dotted blue lines, no marker = upper and lower 95% confidence limits; single red circle = 5% significance level; double red circle = 1% significance level; double red circle with star = 0.1% significance level.

and the last two and first two weeks of November and December, respectively), also attributable to A11 and B03 subgroups.

OTC sales in the M and N ATC anatomical main groups had the highest peak in sales (161% and 191% increase in observed versus predicted, respectively) in the week of March 11th, followed by lower than predicted sales for the remainder of the year. These increases were attributable to sales for OTC medications in the therapeutic subgroup M01 (*anti-inflammatory and antirheumatic products)* and N02 (*analgesics*). Notably, there was a 12% increase in the observed versus predicted sales for the N07 subgroup, of which only N07BA (*drugs used in nicotine dependence*) is sold as OTC. ATC group R had a sustained 41–86% increase in observed versus predicted OTC DDD sales during the three weeks from March 11[th] to the end of March, due to sales for R01 (*nasal preparations*), R02 (*throat preparations*), R05 (*cough and cold preparations*), which subsequently decreased in the observed versus predicted sales for the remainder of the year. The increase in sales in June is attributed to sales of R06 (*antihistamines for systemic use*).

## Discussion

Beginning in mid-February 2020, there was an increase in the number of DDDs of prescription medication dispensed in all ATC anatomical main groups studied, with the peak occurring during the week starting on March 11[th], coinciding with the declaration of the pandemic by the WHO [2]. Overall, there was a 46% increase in the observed versus the predicted amount of DDDs dispensed in this particular week based on data from the five preceding years (approximately 5,000 DDDs per 1000 inhabitants more than expected), with increases of over 100% for some ATC therapeutic subgroups. An increase was found in all age groups, in both men and women, and across both metropolitan and non-metropolitan regions in Sweden. Similarly, in mid-February 2020 weekly OTC sales in DDDs at pharmacies and grocery stores began to increase across Sweden, reaching the peak of 95% above the predicted amount (approximately 1,200 DDDs per 1000 inhabitants) in the week starting on March 11[th].

From the peaks observed in the week starting on March 11[th], DDDs from both prescription dispensations and OTC sales decreased by a fifth in the following week, which included March 19[th] when the government issued the recommendation that pharmacies limit the amount of prescribed and OTC medication sold. By the week beginning on April 1[st], when the restriction on sales was formally implemented, the medication dispensation and sales overall had already decreased to the predicted amount based on seasonal patterns in previous years.

The aggregated data used for this study does not allow us to determine whether the observed increase in DDDs dispensed in March 2020 are due to an increase in the number of new users of medication or to stockpiling behavior of individuals who already had a prescription and were using the medication. However, it can be speculated that the latter scenario is more likely due to the relatively short period of time between the high alert declared by the National Board of Health and Welfare at the end of February and the peak in dispensed DDDs in the second week of March. During this short interval, there was not enough time for large numbers of individuals to visit a doctor to receive a new prescription, and in support of this, it has been reported that the number of primary care visits decreased by approximately 15%

between March and May 2020, even when taking into account the rise in virtual appointments, compared to the average number of visits in January to February 2020 [10].

The contribution of new users versus stockpiling behavior to the increase in prescription dispensations will differ between diagnoses and medications. One possibility is that individuals with chronic disease such as diabetes and cardiovascular disease may have decided to dispense extra supplies of their medication in case a potential societal lock-down prevented them from visiting the pharmacy, reflected in the increase in the DDDs dispensed for antidiabetic medication and cardiovascular system drugs. Relatedly, there were reports in the media concerning potential insulin shortages in Sweden [11]. Further, for many of the ATC groups in the weeks of April to June, we see that the number of DDDs dispensed was in the lower end of the confidence limits, possibly showing that with a larger supply of medication at home due to stockpiling, fewer visits to the pharmacy were needed.

A second possibility is that individuals with conditions that put them at risk of suffering from more severe forms of coronavirus disease 2019 (COVID-19) stockpiled extra medication. We found that the largest increase in DDDs dispensed were for medications used to treat asthma—a condition identified early in 2020 as a possible risk factor for severe forms of COVID-19. It was recommended by some health regulatory agencies worldwide that an individual with asthma ensure that they had at least a full month's supply of medication at home to adequately control asthma and in case of exacerbation due to illness [12], albeit not in Sweden.

A third possibility contributing to the observed increase in DDDs dispensed are medications that were purported to prevent SARS-CoV-2 infection and treatment of COVID-19. For example, use of medications in the ATC group P are relatively low in Sweden, however there was doubling of the DDDs dispensed in March 2020. This ATC group contains the drugs chloroquine and hydroxychloroquine which some studies published in early 2020 showed beneficial effects in patients with COVID-19, however further studies showed no effect on COVID-19 and raised concern over side effects of its use [13]. Both the European Medicines Agency and the Swedish Medicine Products Agency issued restrictions on these drugs in the first week of April 2020 [14,15].

Notably, after a small increase in the week of March 11[th], we show a substantial decrease in the weekly observed versus predicted DDDs dispensed for drugs from the ATC group J throughout the remainder of the year, primarily driven by antibiotic medications. The Swedish National Board of Health and Welfare has reported that this decrease in antibiotic use due to a decrease in the number of severe infections, particularly respiratory infections, in Sweden in 2020, likely due to increased physical distancing and improved hygiene routines during the pandemic [16]. This phenomenon has also been observed in Denmark and Finland which had more stringent social restrictions.

We report a substantial increase in the sales of OTC vitamins and medications used to treat aches and fever, as well as nasal, throat, and cough and cold preparations. This is consistent with the Swedish Medicine Products Agency's report in mid-March that a specific package sizes of paracetamol was backlisted [17] and the Swedish National Board of Health and Welfare's report that two-thirds of the increase in OTC sales in February and March 2020 were due to sales of paracetamol and ibuprofen [5].

The reasons for the increase in OTC sales cannot be determined from the aggregated data used. However, potential reasons include an increase in the number of individuals experiencing and treating symptoms of illness at home. This is supported by the noted reduction in primary care visits in Sweden during the same time period [10]. Additionally, the increase may be a result of household stockpiling of OTC medications in preparation of illness or due to worry of potential drug shortages or lockdown restrictions. The subsequent decrease in the observed

versus predicted sales for the remainder of the year points to sizable amounts of medication stockpiled at the time of purchase in March.

In October 2020, the Swedish National Board of Health and Welfare issued a report on the effect of the COVID-19 pandemic on medication use in the country [5]. They found that in the first weeks of March, medication dispensations from pharmacies increased for drugs used to treat asthma, and to a smaller extent, cardiovascular drugs. For the majority of the most common medications used in Sweden, there was no notable change seen in dispensation patterns in 2020 compared to 2017–2019.

However, their unit of measure was the number of individuals dispensing medication, which does not capture the amount they are dispensing, leading to the differences between their findings and the present study. Their description of increased OTC sales resembles our findings since DDDs were used in both cases.

Similar patterns in medication sales to our study have been reported in neighboring Norway and Finland. The Norwegian Institute of Public Health has reported that sales of OTC DDDs of paracetamol and ibuprofen increased by 6–7% in the first half of 2020 compared to the first half of 2019, with the sharpest increase in March. During the same time period, there was a 4% decrease in nasal sprays for common colds and a 9% increase in DDDs of nicotine cessation products sold [7]. In Finland, an increase in dispensed medications overall and for each ATC anatomical main group is reported for the weeks in March 2020 compared to the same weeks in March 2019, however, the peak is reached in the 3rd week of March, one week after the peak in Sweden. Notably, the Social Insurance Institution of Finland also displays the cumulative amount of medication dispensed from the beginning of 2020, which is the same as the cumulative amount in 2019, indicating that stockpiling, instead of new medication users, is likely behind the March peaks [6].

That three countries report similar phenomena regarding medication dispensing and OTC sales with the arrival of the SARS-CoV-2 virus, suggests that similar behavior should be anticipated at the start of pandemics in the future. Amidst the shortages of critical drugs widely reported by many countries, including Sweden, during the start of the COVID-19 pandemic the discussion among the international community on mitigation strategies was focused primarily on IV sedatives used to facilitate mechanical ventilation [18–20]. It should be acknowledged that stockpiling behaviors by large sections of the population can lead to depletion of national reserves of medications for chronic disease (e.g., diabetes, asthma [21]) as well as treatment of infection symptoms, and calls for consideration within such strategies to prevent shortages in future scenarios.

One strength of this study is the use of nation-wide data for drug prescription dispensing from pharmacies and OTC sales for the entire Swedish population since according to Swedish law, those authorized to conduct wholesale trade in medicines must report information on drug sales each month. The pattern of dispensations over one year shows seasonal variation influenced by holidays in Sweden. Hence, a second strength is the use of data from 2015–2019 to create a prediction model incorporating details of work days, holidays, and seasonal variation. However, the prediction models are extrapolations into the future. The models contains a linear time trend to capture changes in drug use over time, however if the change is non-linear then the extrapolation will be incorrect. We have inspected the predictions for 2015 to 2019 to check the validity of the linear trend. For example, we excluded the P01 OTC sales from analyses as periods of shortages in 2018 and 2019 made extrapolation to 2020 too uncertain.

One limitation is that the use of aggregated data does not allow us to investigate the behavior of individuals and identify new users of medication or increases in treatment dose versus stockpiling behavior. Further, our data does not include medication dispensed by hospital

pharmacies to admitted patients. While data on drug purchasing by hospitals and clinics is available, this focus is not within the scope of this study.

## Conclusions

From the announcement of the high alert in Sweden due to the COVID-19 pandemic in mid-February to mid-March, there were significant increases in the weekly volumes of prescription medication dispensed and OTC medications sold compared to predictions based on data from 2015–2019. Volumes quickly decreased following recommendations from public authorities. Our findings are suggestive of stockpiling over a surge in new users of medication, with variations in the patterns of such behavior between medication classes. In light of similar reports from two neighboring Nordic countries, stockpiling behavior should be anticipated at the start of pandemics in the future.

## Supporting information

**S1 Fig. Volume of dispensed defined daily doses of prescription medication per 1000 inhabitants by week, 2015–2020, Sweden (a) by sex, (b) by region.** Note: The vertical lines indicate the weeks containing March 19th and April 1st 2020 when limits on medication sales were recommended and then mandated, respectively.
(TIF)

**S2 Fig. Volume of dispensed defined daily doses of prescription medication per 1000 inhabitants by week and by ATC anatomical main group, 2015–2020, Sweden.** Note: The vertical lines indicate the weeks containing March 19th and April 1st 2020 when limits on medication sales were recommended and then mandated, respectively.
(TIF)

**S3 Fig. Volume of defined daily doses sold of over-the-counter medications per 1000 inhabitants by week and by ATC anatomical main group, 2015–2020, Sweden.** Note: The vertical lines indicate the weeks containing March 19th and April 1st 2020 when limits on medication sales were recommended and then mandated, respectively.
(TIF)

**S1 Table. Observed versus predicted volume of dispensed defined daily doses of prescription medication per 1000 inhabitants in Sweden for weeks in 2020 with statistically significant differences.**
(PDF)

**S2 Table. Observed versus predicted volume of defined daily doses sold of over-the-counter medications per 1000 inhabitants in Sweden for weeks in 2020 with statistically significant differences.**
(PDF)

## Acknowledgments

The authors thank Anders Ekbom from Clinical Epidemiology Division, Karolinska Institutet for his feedback on the manuscript.

## Author Contributions

**Conceptualization:** Pär Karlsson, Aya Olivia Nakitanda, Lukas Löfling, Carolyn E. Cesta.

**Data curation:** Pär Karlsson, Carolyn E. Cesta.

**Formal analysis:** Pär Karlsson.

**Funding acquisition:** Carolyn E. Cesta.

**Investigation:** Pär Karlsson, Aya Olivia Nakitanda, Lukas Löfling, Carolyn E. Cesta.

**Methodology:** Pär Karlsson.

**Project administration:** Carolyn E. Cesta.

**Resources:** Carolyn E. Cesta.

**Software:** Pär Karlsson.

**Supervision:** Pär Karlsson, Carolyn E. Cesta.

**Validation:** Pär Karlsson, Aya Olivia Nakitanda, Lukas Löfling, Carolyn E. Cesta.

**Visualization:** Pär Karlsson, Aya Olivia Nakitanda, Lukas Löfling, Carolyn E. Cesta.

**Writing – original draft:** Carolyn E. Cesta.

**Writing – review & editing:** Pär Karlsson, Aya Olivia Nakitanda, Lukas Löfling, Carolyn E. Cesta.

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
