## [Decision Letter · Decision Letter 0]

20 May 2021

PONE-D-21-06619

Patterns of prescription dispensation and over-the-counter medication sales in Sweden during the COVID-19 pandemic

PLOS ONE

Dear Dr. Cesta,

Thank you for submitting your manuscript to PLOS ONE. After careful consideration, we feel that it has merit but does not fully meet PLOS ONE’s publication criteria as it currently stands. Therefore, we invite you to submit a revised version of the manuscript that addresses the points raised during the review process.

We look forward to receiving your revised manuscript.

Kind regards,

Sherief Ghozy, M.D., Ph.D. candidate

Academic Editor

PLOS ONE

Journal Requirements:

Reviewers' comments:

Reviewer's Responses to Questions

**Comments to the Author**

1. Is the manuscript technically sound, and do the data support the conclusions?

Reviewer #1: Yes

Reviewer #2: Yes

2. Has the statistical analysis been performed appropriately and rigorously? 

Reviewer #1: I Don't Know

Reviewer #2: Yes

3. Have the authors made all data underlying the findings in their manuscript fully available?

Reviewer #1: Yes

Reviewer #2: Yes

4. Is the manuscript presented in an intelligible fashion and written in standard English?

Reviewer #1: Yes

Reviewer #2: Yes

5. Review Comments to the Author

Reviewer #1: The study is relevant and well conducted. Here are just some minor comments that might slightly improve this well-written manuscript.

The introduction is very brief. It would be useful to the reader if eg. more details of the preventive actions taken by Läkemedelsverket were given.

107 Consider replacing the word “interventions” with “products” or similar.

266 Individuals normally don’t dispense themselves, they tend to have their medication dispensed.

355 Please include the reference.

Reviewer #2: 1-List abbreviation after full words such as (SARS-CoV-2)

2-No ethical approval was obtained?

3-In discussion section, please add a paragraph about the increase of OTC medication due to the fear of shortage of drugs due to lockdown, people buy a large amount of OTC medication at one time for sticking to the predicted lockdown which occurred in the last of March 2020,....etc

4-Was there an overwhelmed hospitals and clinics during this surge of OTC medication? Because it can be another explanation to that surge as people will try to treat symptoms at home.

6. PLOS authors have the option to publish the peer review history of their article (what does this mean?). If published, this will include your full peer review and any attached files.

Reviewer #1: No

Reviewer #2: No

---

## [Author Response · Author response to Decision Letter 0]

10 Jun 2021

PONE-D-21-06619

Patterns of prescription dispensation and over-the-counter medication sales in Sweden during the COVID-19 pandemic

PLOS ONE

Dear PLOS ONE Academic Editor Sherief Ghozy and reviewers, 

Thank you for your thoughtful review of our manuscript and for the opportunity to submit a revised version for your consideration. 

Two copies of the manuscript are submitted, one a clean copy, the other with tracked changes. Any mention of page numbers in the point-by-point response refer to the clean version.

Note that we have initiated some revisions because in the time since submitting the manuscript, we have acquired data for the remainder of 2020 (October to December). All figures and tables have been updated with the additional data. The methods, results and discussion has been updated accordingly. The notable new results include a lower than predicted volume of dispensations of ATC group J until the end of the year, and substantial decrease in OTC drugs in ATC group B at the end of 2020, otherwise the main findings presented in the first submission remain the same.

Sincerely,

Carolyn Cesta

Author’s response to reviewer comments (in blue)

Reviewer #1: 

The study is relevant and well conducted. Here are just some minor comments that might slightly improve this well-written manuscript.

The introduction is very brief. It would be useful to the reader if eg. more details of the preventive actions taken by Läkemedelsverket were given.

We have moved from Discussion to the Introduction section (lines 92-96) details about prescription filling in Sweden and the restrictions on them by Läkemedelsverket. 

107 Consider replacing the word “interventions” with “products” or similar.

As recommended, “interventions” has been changed to “products”

266 Individuals normally don’t dispense themselves, they tend to have their medication dispensed.

The sentence has been rephrased to “The contribution of new users versus stockpiling behavior to the increase in prescription dispensations will differ between diagnoses and medications”(line 267 in clean version)

355 Please include the reference.

Since we now have complete data for all of 2020, these sentences have been removed from the limitations in the discussion. 

Reviewer #2: 

1-List abbreviation after full words such as (SARS-CoV-2)

The full name of SARS-CoV-2 and COVID-19 has now been added before their abbreviations when first mentioned in the abstract and the main text.

2-No ethical approval was obtained?

Ethical approval was not required as the data requested from the register holders was provided as 

aggregated number of DDDs and not as individual level dispensation data. The Ethical Review Act (2003:460) on ethical review of research involving humans in Sweden dictates that ethical approval is needed when processing sensitive personal data, however data in the aggregated form does not allow for the identification of individuals and does not require ethical approval. 

3-In discussion section, please add a paragraph about the increase of OTC medication due to the fear of shortage of drugs due to lockdown, people buy a large amount of OTC medication at one time for sticking to the predicted lockdown which occurred in the last of March 2020,....etc

4-Was there an overwhelmed hospitals and clinics during this surge of OTC medication? Because it can be another explanation to that surge as people will try to treat symptoms at home.

To address points 3 and 4 by reviewer 2, we have re-written the paragraph about potential reasons why there was an increase in OTC medication sales in March 2020 in lines 304-310 in the clean version: 

The reasons for the increase in OTC sales cannot be determined from the aggregated data available. However, potential reasons include an increase in the number of individuals experiencing and treating symptoms of illness at home. This is supported by the noted reduction in primary care visits in Sweden during the same time period.[10] Additionally, the increase may be a result of household stockpiling of OTC medications in preparation of illness or due to worry of potential drug shortages or lockdown restrictions. The subsequent decrease in the observed versus predicted sales for the remainder of the year points to sizable amounts of medication stockpiled at the time of purchase in March.

---

## [Editor Report · Decision Letter 1]

16 Jun 2021

Patterns of prescription dispensation and over-the-counter medication sales in Sweden during the COVID-19 pandemic

PONE-D-21-06619R1

Dear Dr. Cesta,

We’re pleased to inform you that your manuscript has been judged scientifically suitable for publication and will be formally accepted for publication once it meets all outstanding technical requirements.

Kind regards,

Sherief Ghozy, M.D., Ph.D. candidate

Academic Editor

PLOS ONE

---

## [Editor Report · Acceptance letter]

29 Jul 2021

PONE-D-21-06619R1 

Patterns of prescription dispensation and over-the-counter medication sales in Sweden during the COVID-19 pandemic 

Dear Dr. Cesta:

I'm pleased to inform you that your manuscript has been deemed suitable for publication in PLOS ONE. Congratulations! Your manuscript is now with our production department. 

Kind regards, 

on behalf of

Dr. Sherief Ghozy 

Academic Editor

PLOS ONE